# Online Learning-Based Hybrid Tracking Method for Unmanned Aerial Vehicles

**DOI:** 10.3390/s23063270

**Published:** 2023-03-20

**Authors:** Sohee Son, Injae Lee, Jihun Cha, Haechul Choi

**Affiliations:** 1Department of Multimedia Engineering, Hanbat National University, Daejeon 34158, Republic of Korea; 2Electronics and Telecommunications Research Institute, Daejeon 34129, Republic of Korea

**Keywords:** computer vision, drone, unmanned aerial vehicles, object tracking, object detection

## Abstract

Tracking unmanned aerial vehicles (UAVs) in outdoor scenes poses significant challenges due to their dynamic motion, diverse sizes, and changes in appearance. This paper proposes an efficient hybrid tracking method for UAVs, comprising a detector, tracker, and integrator. The integrator combines detection and tracking, and updates the target’s features online while tracking, thereby addressing the aforementioned challenges. The online update mechanism ensures robust tracking by handling object deformation, diverse types of UAVs, and changes in background. We conducted experiments on custom and public UAV datasets to train the deep learning-based detector and evaluate the tracking methods, including the commonly used UAV123 and UAVL datasets, to demonstrate generalizability. The experimental results show the effectiveness and robustness of our proposed method under challenging conditions, such as out-of-view and low-resolution scenarios, and demonstrate its performance in UAV detection tasks.

## 1. Introduction

With the recent development of drone technologies, the practical applications of unmanned aerial vehicles (UAVs), also known as drones, are becoming increasingly diversified [1]. However, the diversity of using drones can incur negative effects, since drones have multidirectional purposes. For example, unauthorized use of UAVs, e.g., hovering over airports, military facilities, and private compounds, can pose threats related to security and privacy. An accident at these locations could lead to serious disasters, endangering both human life and property. To mitigate the adverse effects of such incidents, governments regulate UAV operations through civil aeronautics laws. However, it is challenging to restrict all uncontrollable UAVs [1]. To address the unauthorized use of UAVs, various systems using radar, radio frequency signals, and images to detect, identify, and track illegal UAVs have been actively investigated [2]. This paper focuses on the issue of single object tracking in images captured from the ground or sky to ensure the surveillance of unauthorized UAVs.

Visual object tracking in images is a fundamental computer vision task, e.g., automatic driving, augmented reality, and visual surveillance. Despite significant advancements in deep learning-based object tracking in recent years, there are still several challenges that need to be addressed, such as dealing with distractors caused by similar objects, and changes in the appearance of the tracked object [3,4]. In addition, tracking can easily drift to the background due to occlusions, target objects that are out of view, and the abrupt appearance of objects. The small size of objects makes tracking extremely difficult because of the limited amount of available visual information. Tracking UAVs in outdoor environments is particularly challenging for the following reasons: (1) wide-area surveillance, including the sky and ground, because of UAVs’ wide range of activities; (2) the diverse sizes and appearances of UAVs; (3) low image resolution because moving objects can occupy a small area; (4) significant distance between the camera and the moving objects; (5) large interframe disparity caused by fast UAV and camera motion; and (6) uncontrollable environmental conditions, e.g., illumination changes and weather conditions.

Many tracking methods have been proposed in recent years to handle these challenging problems. For example, one approach integrates tracking and detection, where the key is how to determine the output from the tracker and detector. Wang et al. [5] combined the tracker and detector with a linear combination. In addition, Saribas et al. [6] proposed a mechanism to switch between the tracker and detector. This mechanism compares the Euclidean distances and intersection-over-union (IoU) ratio between the detected bounding boxes and the latest output of the tracker. However, these integration methods [5,6] focus on the location and size of the bounding boxes obtained from the tracker and detector, rather than the target’s appearance. As a result, these methods are limited in their ability to accommodate object deformation. Baptista et al. [7] proposed a surveillance system in which the deep convolutional ResNet-50 model performs object classification on tracked aerial objects to detect aerial targets. Although the pretrained classifier realizes reasonable performance in terms of tracking and classifying the target, the pretrained model may be limited in terms of generalizability due to a lack of datasets for different types of UAVs. Rozantsev et al. [8] proposed convolutional neural network-based regressors on spatiotemporal image cubes extracted using a multi-scale sliding window to detect flying objects. They demonstrated the robust performance of changes in the appearance of flying objects with low resolution, but the entire procedure could not be executed in real-time. 

Thus, this paper proposes an online learning-based hybrid tracking method that combines detection and tracking to overcome the above challenges, and enables trackers to be more discriminative for flying object tracking. Object tracking is pertinent for localizing an object of interest by exploiting the object correspondence between frames; however, distractors, e.g., similar looking objects and appearance changes, can cause the tracker to shift focus from the target to the background. The proposed tracking method addresses this issue by integrating object detection to determine the locations and scales of objects in an image. However, this demands perceiving detections that could potentially belong to distractors because they are unfavorable to differentiate the target object from other objects in the same category. Thus, the proposed method integrates both a detector and a tracker by predicting the probability of the object’s presence using an online learning classifier. The classifier attempts to mitigate the limitations of the tracker and detector, while exploiting the benefits of each. In addition, the classifier updates the features in an online manner rather than prior learning, which improves robustness against abrupt changes in an object’s appearance. As a result, the proposed hybrid tracking method exhibits powerful localization capability by handling both mistracking and misdetection. 

In particular, the main contributions of this paper can be summarized as follows: (1) This paper introduces a robust drone-tracking approach that integrates detecting and tracking algorithms based on an online learning classifier. (2) This paper evaluates the proposed method for the two main goals, which are tracking and detection, on our own drone dataset and drone-vs-bird dataset. The proposed method outperforms existing trackers. (3) This paper performs an evaluation on common aerial video datasets, which are UAV123 and UAVL, to demonstrate the generalization.

The remainder of this paper is organized as follows. Section 2 introduces work related to object detection and tracking. Section 3 describes the proposed hybrid tracking method, including the online learning-based integration method. Section 4 discusses experiments conducted on drone-based datasets to evaluate the proposed method. Finally, the paper is concluded in Section 5.

## 2. Related Work

In the following, we introduce conventional detection and tracking methods.

### 2.1. Object Detection

Deep learning techniques, which have emerged as a powerful way to automatically learn feature representations from data, have realized important improvements in the object detection field [9]. Existing domain-specific image object detectors can generally be divided into two categories [10], i.e., two-stage detectors, e.g., Faster R-CNN [11], and one-stage detectors, e.g., YOLO [12].

Two-stage object detectors generate category-independent region proposals, and classify the proposals in a category-specific manner [9]. In a pioneering work, Girshick et al. [13] explored CNNs for generic object detection, and they proposed the R-CNN, which is a three-stage pipeline detection system. They created region proposals that may contain objects via a selective search process and extracted features from each region. Then, they classified each region into an object category using a set of class-specific linear SVMs. Despite providing outstanding contributions to the object detection field, the R-CNN method has several prominent drawbacks, e.g., inefficient individual training of multiple stages, high training costs, and slow processing time [14]. Inspired by this, the Fast R-CNN [14] streamlined the training process by jointly learning to classify proposals and refine their bounding boxes. Here, the key is to share convolution computation across proposals and add a region of interest (RoI) pooling layer in order to extract a fixed-length feature vector from the feature map. Although the Fast R-CNN method improves the speed of the detection process, a bottleneck exists on the external region [11]. Faster R-CNN [11] applies the region proposal network (RPN) to the Fast R-CNN technique to realize efficient and accurate region proposal generation, which enables more efficient computation. With flexible and accurate performance for region-based classification, two-stage object detectors have progressed, e.g., FPN [15], R-FCN [16], and Mask R-CNN [17]; however, these methods have also demonstrated high computational costs [10].

Unlike two-stage detectors, one-stage detectors, which do not separate the region proposal process, predict object categories and bounding box offsets directly for the entire image using a single feedforward CNN [9]. This results in a considerable speed increase at the expense of relatively lower accuracy. YOLO [12], the most representative model, treats object detection as a regression problem to spatially separate bounding boxes and the associated class probabilities. YOLO predicts the bounding boxes directly from an image gridded into several regions. This unified design enables end-to-end training and real-time processing at 45 frames per second (fps) [12]. Redmon and Farhadi [18] proposed YOLOv2, which is an improved version of YOLO. YOLOv2 offers improved accuracy by applying considerable skill and experience from other studies, faster speed by replacing the feature extractor with Darknet-19, and better performance that can detect over 9000 object categories. Similarly, Redmon and Farhadi introduced YOLOv3 [19] by incorporating various techniques, including the feature extractor network of DarkNet-53, making YOLOv3 more accurate but still fast. Bochkovskiy et al. [20] proposed YOLOv4. Their main goal was to design a fast and easy-to-train object detector in production systems and optimize parallel computations. Unlike most existing approaches requiring multiple GPUs for training, this method can train using only a single GPU, while maintaining high speed and accuracy. Although one-stage detectors generally produce lower detection accuracy than two-stage detectors, this more straightforward unified pipeline strategy is promising because such techniques can run in real-time at acceptable memory costs; thus, one-stage detectors have attracted more attention in terms of real-world applications.

Although deep learning-based object detection techniques have achieved great progress, detecting very small objects, e.g., UAVs, remains challenging. Aker and Kalkan [21] created an artificial dataset to fine-tune YOLOv2 to detect UAVs by collecting public domain pictures of UAVs and birds, and then combining background-subtracted authentic images. To detect flying objects from a moving camera, Rozantsev et al. [8] proposed the detection pipeline based on the concept of motion compensation and classification of the spatiotemporal image cubes (st-cubes). Here, CNN-based regressors are applied to realize motion compensation on st-cubes extracted using a multi-scale sliding window approach, and then each st-cube is classified as containing an object of interest or not, using the CNN-based classifier. The authors demonstrated the potential of CNN-based detection for flying objects with low resolution. However, this pipeline may not be sufficient for real-time surveillance systems due to its computational costs.

### 2.2. Object Tracking

Currently, popular visual tracking methods can generally be divided into two branches. The first branch is based on online learning, which learns the features of an object of interest while tracking. Representative trackers include the kernelized correlation filter (KCF) [22] and boosting [23]. The KCF is a well-known correlation filter-based method that discriminates the target object from the background in the Fourier domain. Boosting selects features to differentiate the object from other objects using an online boosting classifier and updates the classifier whenever a new training sample is available for visual tracking. Online learning-based trackers are robust against distractors through effective appearance adaptation during tracking; however, the online adaptation could exacerbate the quality of the template. Thus, the tracker should have the discriminating capability to prevent pollution from the poor quality of the template.

Another branch of tracking methods implements offline learning, which exploits models pretrained on vast datasets to extract features. Among other things, Siamese neural networks have received increasing attention in the object tracking field due to their adequate balance between performance and processing speed [24]. Siamese trackers attempt to find an object in specified search regions using a learned similarity function that identifies whether the object is present [4]. The pioneering representative work is the fully convolutional Siamese network (SiamFC) [25]. In the SiamFC architecture, a cross-correlation layer fuses the target and search templates to produce the response map of the similarity to localize the object. One advantage of this architecture is that it enables computation of the similarity at all sub-windows of the search template in a single pass. The succinct SiamFC architecture has demonstrated effective and powerful performance, and follow-up studies have primarily focused on Siamese networks, including SiamVGG [26], DSiam [27], SA-Siam [28], SiamRPN [29], SiamRPN++ [30], DaSiamRPN [31], SiamCAR [32], SiamMask [33], SiamBAN [34], and SE-SiamFC [35]. Even though offline learning-based trackers exhibit high robustness and speed, they still face tracking drift problems caused by occlusions, out-of-view, and object deformation.

As described in Section 1, tracking small flying objects, e.g., UAVs, is challenging due to insufficient availability of features to inform the object, and large interframe disparity to surveil rapid objects. When the tracker suffers from problems caused by these issues, a mechanism similar to redetection can realize significant improvements. Therefore, an approach that uses detection and tracking together has been studied to improve robustness of the tracker. For example, Saribas et al. [6] exploited YOLOv3 to detect a target object in the first frame and recover tracking failures, and they used the KCF to track the object. This mechanism relied on the confidence score of the KCF to switch between the detector and tracker. When the score is under a given threshold, the target’s new position is selected according to Euclidean distances and the IoU ratio between candidates from the detector and the tracker’s most recent output. In addition, Cintas et al. [36] utilized YOLOv3-Tiny and the KCF, and they activate a switch mechanism every 30 frames, or in cases where the tracker fails. Then, based on the Euclidean distance between the predicted results by the detector and the latest result produced by the tracker, the result with the closest location is considered the target. Wang et al. [5] proposed a hybrid detector–tracker for birds and nests. Here, if the disagreement measure derived by bounding boxes from the detector and tracker is less than a given threshold, the target is localized by a linear combination of the detector and tracker. If the disagreement measure satisfies the threshold, the bounding box of the detector is determined to be the target. Although the approaches in [5,6,36] are fast and straightforward, the target is localized using only the bounding box (rather than appearance features); thus, this approach is ineffective at handling the tracking drift problems. Therefore, a sophisticated mechanism is required to track UAVs in real-world situations.

## 3. Methodology

### 3.1. Proposed Framework

In this paper, an online learning-based hybrid tracking method that combines detection and tracking is proposed to track UAVs effectively. Object tracking localizes the target according to the object correspondence between frames; however, the tracking drift problem frequently results in losing the target object. In addition, object detection, which localizes and classifies objects, has limitations relative to distinguishing the target from other objects in the same class, and handling the diverse sizes and appearances of UAVs. As discussed in Section 1, tracking a flying object, e.g., UAVs, in outdoor environments is challenging. To recover tracking when the tracker loses the target object, the proposed hybrid tracking method integrates both a detector and tracker. Here, the key is to derive the output by predicting the probability of objectness, which decides the likelihood of an object, via the online learning classifier. A powerful feature that could serve for discriminating the object from others is required in achieving robust tracking; thus, the classifier learns the object’s features during tracking to accommodate changes in the object’s appearance. This improves the ability to discriminate misdetection and mistracking, and it realizes more robust performance.

Figure 1 shows the framework of the proposed hybrid tracking method. The proposed method includes three main components, i.e., the detector (Section 2.1), the tracker (Section 2.2), and the integrator, which is used to learn the object’s features, predict their confidences, and switch the detector and tracker. The tracking process begins after the framework is initialized when an object of interest is given. First, the detector and tracker each predict a bounding box in a given image. To realize an effective balance between accuracy and processing speed, the detector is employed to locate objects either at regular frame intervals or in cases where the tracker fails. The controller is used to send each on/off signal associated with the execution to the detector and tracker. For convenience, the output bounding boxes of the detector, tracker, and integrator are expressed as follows:(1)$$B_{i}=\left[x_{1}^{B_{i}},y_{1}^{B_{i}},x_{2}^{B_{i}},y_{2}^{B_{i}}\right],(i=D,T,and\;O),$$
where $\left(x_{1}^{B_{i}},y_{1}^{B_{i}}\right)$ and $\left(x_{2}^{B_{i}},y_{2}^{B_{i}}\right)$ indicate the top-left and bottom-right coordinates of the bounding box, respectively. With $B_{D}$ and $B_{T}$ given by the detector and tracker, respectively, the integrator derives the result $B_{O}$ from the parameters of the strong classifier passed by the updater. Here, if the detector and tracker do not predict the target or perform by the passed on/off signal from the controller, the input to the integrator can be either $B_{D}$ or $B_{T}$. To draw the output from $B_{D}$ and $B_{T}$ in the integrator, the strong classifier predicts ${conf}_{D}$ and ${conf}_{T}$, which are the confidences of $B_{D}$ and $B_{T}$, respectively. The confidence score reflects the objectness score that determines the likelihood of an object. With the confidence score, the target decision module derives the output $B_{O}$ based on a given threshold, which attempts to circumvent obstacles by discerning misdetection and mistracking, as described in the following two scenarios. In the first scenario, if one of $B_{D}$ and $B_{T}$, whichever has the larger confidence score, is above the threshold, the bounding box associated with this confidence becomes the output $B_{O}$. The output is then passed to the updater for online training of the parameters of the strong classifier, and to the tracker for the subsequent tracking process. For training, the patch generation collects training samples based on the output $B_{O}$ as the positive sample, and the surrounding regions represent negative samples. The details of the updater are discussed in Section 3.2. 

In the second scenario, if the larger confidence score, either $B_{D}$ or $B_{T}$, is unsatisfied based on the threshold, the target decision module considers all bounding boxes as the object’s absence, and then discards all bounding boxes.

The proposed hybrid tracking method can deploy diverse detection and tracking algorithms depending on the specific goals and circumstances of the corresponding application. For example, to detect small objects, e.g., UAVs, EfficientDet [37] and FPN [15] can be employed to extract semantically strong features by fusing multi-scale features. YOLO [12] and SSD [38] can benefit from their efficient and fast approaches. For the tracking, Siamese-based trackers can yield effective and powerful performance; however, non-deep learning algorithms, e.g., MedianFlow [39] and the KCF [22], are suitable for real-time tracking on onboard computers with limited hardware resources.

### 3.2. Online Classifier for Integration

In the proposed method, the online learning-based AdaBoost classifier is employed to integrate the detector and tracker, thereby mitigating their drawbacks by predicting their corresponding confidences. The AdaBoost algorithm has been applied in a wide range of machine learning tasks and applications as an ensemble algorithm [40]. AdaBoost builds a strong classifier as a linear combination of weak classifiers. Based on an online boosting approach [41], a variety of computer vision applications have been researched, e.g., object detection [42] and visual object tracking [40,43]. The core concept of the online boosting approach is that it updates all weak classifiers using a single training sample compared to offline boosting, which updates a single weak classifier using all available training samples. Based on [41], Grabner and Bischof [40] proposed the online AdaBoost method for feature selection using a strong classifier comprising selectors rather than a weak classifier. Here, each selector holds a weak classifier corresponding to the global feature pool. The primary purpose of the selector is exploiting online boosting to the selectors, not directly to the weak classifiers. They demonstrated that this technique realizes efficient computation of features during tracking, and allows the tracker to be performed in real-time. Inspired by [40,43], the proposed method implements an integrator with the online AdaBoost classifier to reduce the negative impact of the distractors.

The proposed hybrid tracking method classifies the bounding boxes from the detector and tracker by exploiting the parameters of a strong classifier passed by the updater, which is trained using samples generated by the previous tracking result. The strong classifier’s confidence is measured by the linear combination of selectors as follows:(2)$$conf\left(x\right)=\sum_{n=1}^{N}\alpha_{n}\cdot h_{n}^{sel}\left(x\right)$$
where $h_{n}^{sel}$ and $\alpha_{n}$ are n-th selector and its voting weight, respectively, given a set of N selectors, and x is the input patch by $B_{D}$ and $B_{T}$. Training the updater means that weak classifiers are updated, and each selector chooses the best weak classifier with the lowest estimated error.

In the patch generation module (Figure 1), the training samples $X=\left\{B_{1},B_{2},\ldots ,B_{L}\right\}$ are built for online adaptation of the classifier during tracking by exploiting $B_{O}$ as positive sample $B_{1}$, and extracting a set of L−1 negative samples surrounding $B_{O}$. Here, the l-th negative sample $B_{l}$ $\left(l=2,3\ldots ,L\right)$ is defined as follows:(3)$$B_{l}=\left[\begin{array}{cc} x_{1}^{B_{o}}+\alpha_{1}^{B_{l}} & y_{1}^{B_{o}}+\beta_{1}^{B_{l}}\\ x_{2}^{B_{o}}+\alpha_{2}^{B_{l}} & y_{2}^{B_{o}}+\beta_{2}^{B_{l}} \end{array}\right]$$
where $\left(x_{1}^{B_{o}},y_{1}^{B_{o}},x_{2}^{B_{o}},y_{2}^{B_{o}}\right)$ represent the coordinates of the bounding box of $B_{O}$, and $\left(\alpha_{1}^{B_{l}},\beta_{1}^{B_{l}},\alpha_{2}^{B_{l}},\beta_{2}^{B_{l}}\right)$ represent the offsets of $B_{l}$.

The algorithm for the training selectors’ procedure, which constitutes the strong classifier within the updater, is presented in Algorithm 1. The procedure is founded on online learning-based Adaboost [40,43]. For further details regarding the symbols and notations used in Algorithm 1, please refer to [40,43]. Note that the procedure of the training selectors updates the weak classifiers, selectors, and voting weights for classifiers. First, the importance weight λ of a training sample is initialized. Given training sample $B_{l}$ and its corresponding label $y_{l}\in \left\{-1,1\right\}$ (a negative and positive sample, respectively), the set of M weak classifiers of the global feature pool is updated (lines 2–4). Here, each weak classifier classifies the sample and then updates its probability distributions for positive labeled samples and negative labeled samples, depending on the estimated label.

The selectors are updated as in lines 5–29. Given the responses of the weak classifiers hweakBl, the n-th selector hnsel chooses the weak classifier hm+weak, which has the lowest error em (lines 6–16). λmc and λmw represent the number of times that samples are classified correctly and incorrectly, respectively. The error is estimated from λmc and λmw. In accordance with the assumption of the boosting algorithm for binary classification, which stipulates that the error rate must be less than 50%, selectors exhibiting an error rate exceeding 50% are excluded from updates (line 17). Based on the error, the selector’ voting weight αn and the importance weight λ corresponding to the training sample are updated (lines 20–25). To adapt to changes in the object’s appearance and improve classification accuracy, the weak classifier with the highest error hm−weak is initialized and replaced with a new randomly selected weak classifier. All selectors and corresponding weights are updated sequentially with the importance weight of the training sample (lines 26–28). As a result, the learned updater makes the classifier robust against object deformation between frames. In addition, the drift problems can be avoided by filtering out misdetections and mistracking. Here, Haar-like features [43] are used to generate weak hypotheses. Haar-like features can be obtained at low computational complexity using integral images as data structures, which helps realize real-time target classification during tracking [44].
** Algorithm 1:** Online adapting classifier ** Require**: training sample , 〈Bl,yl〉, yl∈{−1,+1} 1:    Set λ=1 2:    **for** m=1,2,…,M**do** 3:       hmweak=update(hmweak,〈Bl,yl〉) 4:    **end for** 5:    **for** n=1,2,…,N **do** 6:      **for** m=1,2,…,M**do** 7:        **if** hmweakBl=yl **then** 8:          λmc=λmc+λ 9:        **else**10:          λmw=λmw+λ11:        **end if**12:        em=λmwλmc+λmw13:      **end for**14:      m+=argminm(em)15:      en=em+16:      hnsel=hm+weak17:      **if** en=0 or en>12 **then**18:        exit19:      **end if**20:      αn=12·ln⁡(1−enen)21:      **if** hmweakBl=yl **then**22:        λ=λ·12·(1−en)23:      **else**24:        λ=λ·12·en25:      **end if**26:    **end for**27:    m−=argmaxm(em)28:    λm−c=λm−w=129:    get new hm−weak


## 4. Experiments and Results

### 4.1. Database

Most research based on supervised learning requires vast datasets with labeled ground truth data [45]. However, there is a lack of available datasets for UAVs, unlike more general objects, e.g., humans and vehicles. Thus, in this study, we constructed a unique dataset to track UAVs from outdoor environments. In addition, we also used the public drone-vs-bird dataset [46,47] to improve the proposed generalizability.

To construct our UAV dataset, we recorded videos of UAVs flying. Here, to ensure data diversity, the videos were captured according to three conditions (Figure 2), i.e., diverse backgrounds (cloud, buildings, mountain, etc.), drone types (Figure 3), and camera types (handheld, pan-tilt-zoom camera, etc.). The constructed UAV dataset contains 360 sequences with spatial resolution greater than or equal to full high definition (FHD). The dataset reflects many real-world challenges, including occlusion (OC), out-of-view (OV), background clutter (BC), camera motion (CM), low resolution (LR), scale variation (SV), and fast motion (FM), as described in Table 1. The dataset was created as part of a funded project, but there is a licensing issue with it. Unfortunately, the dataset is not public.

We used an image dataset to train the detector model. This image dataset was taken from our constructed dataset and was refined by obtaining a drone image per second to reduce overfitting, improve accuracy, and improve generalizability. This dataset, which was used as a training set and a testing set, includes 44,986 images from our constructed dataset, and 94,874 images from the drone-vs-bird dataset.

Table 2 shows a dataset for testing the tracking performance. This dataset comprises six videos (10,232 frames) from the drone-vs-bird dataset and six videos (18,278 frames) from our constructed dataset. Each video sequence is over 25 FPS and annotated with the seven attributes described in Table 1.

### 4.2. Experimental Environment

#### 4.2.1. Evaluation Metrics

We evaluated the proposed method in terms of both tracking and detection. For the tracking evaluation, we used metrics from the Object Tracking Benchmark (OTB) (both 2013 [48] and 2015 [3] versions) and the Anti-UAV Benchmark [49]. The detection performance was compared using the following metrics from the PASCAL VOC [50] and ImageNet challenge [51].

IoU: The IoU is a measure of relative overlap between two bounding boxes. For example, if a tracked bounding box $r_{t}$ and ground truth bounding box $r_{o}$ of a target object are given, their IoU is defined as follows:


(4)
$$\mathrm{IoU}\left(r_{t},r_{o}\right)=\frac{\left|r_{t}\cap r_{o}\right|}{\left|r_{t}\cup r_{o}\right|}$$


Center location error (CLE): The CLE is the Euclidean distance between a tracked center location and a manually labeled ground truth position.

For quantitative analysis, the most common evaluation metrics are precision and the success rate. In the precision plot, a frame is marked as being tracked successfully if the CLE score is less than a given threshold [48]. The success plot marks a successful frame if the IoU score is greater than a given threshold. The precision and success plots give the ratio of successful frames at the specified thresholds, and each plot is delineated by varying the threshold values [4]. In addition, the precision and success plots generally rank the tracking methods at thresholds equal to 20 and 0.5, respectively [48].

In real-world scenarios, there is an increased risk that the tracker drifts to the background caused by primary problems, such as occlusion and out-of-view [4]. However, when a tracking algorithm loses track of the target object, the output can be random, which causes incorrect evaluation of tracking performance [3]. To address these issues, the mean state accuracy (mSA) [49] was used in our evaluations. The state accuracy SA is measured for a given sequence as follows:(5)$$\mathrm{SA}=\sum_{t}^{T}\frac{{IOU}_{t}\times \delta \left(v_{t}>0\right)+p_{t}\times (1-\delta \left(v_{t}>0\right))}{T},$$
where ${IoU}_{t}$ and $v_{t}$ are the IoU and ground truth visibility flag at frame t, respectively. If the target exists in frame t, $\delta \left(v_{t}>0\right)=1$; otherwise, $\delta \left(v_{t}>0\right)=0$. If the tracker predicts that the target is absent, the $p_{t}$ value will be 1; otherwise, the $p_{t}$ value will be 0. The mSA is taken as the average SA value for all video sequences.

Recall that the proposed method employs a detector; thus, it can also be used for the detection task without requiring any adaptation at test time. Here, to evaluate detection performance, we used the F-measure, which is the harmonic mean of precision and recall [52]. Based on the true positive (TP), false positive (FP), and false negative (FN), precision, i.e., the percentage of correct positive predictions, indicates a model’s ability to recognize only relevant objects. Recall that the percentage of correct positive predictions among all given ground truths represents the model’s ability to discern all relevant cases [53]. The measurements classify a detection as correct or incorrect by comparing the IoU. In this study, the IoU threshold was set to 0.3. Details of the metrics are introduced in [24,52,53].

#### 4.2.2. Implementation Details

In these experiments, the proposed method is implemented employing YOLOv4 [20] and MedianFlow [39] as the detector and tracker, respectively, to demonstrate performance that is suitable for surveillance systems, which commonly use low-resource onboard computers. MedianFlow executes at high speed on a single CPU, and YOLOv4 runs in real-time on a single GPU while obtaining high accuracy. It should be noted that the proposed hybrid framework is not restricted to the utilization of a particular detector and tracker. In the proposed method, the online learning-based strong classifier consists of 50 selectors, each with a feature pool of 10 weak classifiers. Images with 614 × 614 are fed to YOLOv4. We train YOLOv4 from scratch for 100 epochs with a batch size of 24 using the image dataset described in Section 4.1. Here, the dataset is randomly composed of 109,360 bounding boxes as the training set, and 30,500 bounding boxes as the testing set.

### 4.3. Experimental Comparison

In order to evaluate the performance of the proposed integration approach, we conducted a comparative analysis of its tracking and detection capabilities with those of MedianFlow [39], YOLOv4 [20], SiamRPN++ [30] (a deep learning-based tracking model renowned for its superior tracking accuracy), and a linear combination-based integration method [5], which we henceforth refer to as LC. Specifically, we compared the proposed approach against YOLOv4 and MedianFlow, owing to their widespread use in the literature. Additionally, we evaluated the performance of our method against SiamRPN++, which serves as a representative example of deep learning-based tracking models. Finally, we included LC in our analysis given its similarity in approach to our proposed method. This comparative analysis also demonstrates the effectiveness of our approach in enhancing the individual performance of commonly used detector and tracker components.

#### 4.3.1. Evaluation for Object Tracking

To report the tracking algorithms’ performance in precision and success plots by varying the thresholds, the one-pass evaluation (OPE) technique was employed in this study. 

The OPE technique runs an algorithm with initialization from the ground truth in the first frame and reports the average precision or success rate of all results [3,47]. Following the evaluation protocol of OPE [3], Figure 4 shows that the proposed method with MedianFlow and YOLOv4 achieved the best success score of 0.747 at a threshold value of 0.3. Thus, the proposed method outperformed the LC [5] (0.719) and YOLOv4 (0.618) methods. In addition, the proposed method produced the best precision score of 0.797 at a threshold value of 20. Compared to the LC method, the proposed method realized a 2.8% improvement in terms of overlap, and a 2.9% improvement in terms of precision. YOLOv4 obtained the best performance when the threshold in the success plot was greater than approximately 0.6, and when the threshold in the precision plot was less than 5. These results indicate that YOLOv4 most accurately predicted the bounding box. Russakovsky et al. reported that humans have difficulty distinguishing an IoU of 0.3 from 0.5 [54]. Accordingly, if the IoU threshold value of 0.3 is considered sufficient to realize steady tracking, tracking methods that incorporate both detection and tracking, e.g., the proposed method and LC method, will exhibit robust performance. In particular, the proposed method is more robust than the LC method, which only uses the location of the bounding box.

As mentioned previously, a tracker can easily drift to the background due to real-world challenges, e.g., occlusion and out-of-view problems; thus, it is important to determine whether the object is, in fact, present. To evaluate this, we used the mSA metric. Table 3 shows that the proposed method achieved the best mSA score of 0.553, which indicates that the proposed method belongs to the top three in terms of SA for all test sequences. The proposed method was able to perceive the object’s absence. Even though the SiamRPN++ method obtained the best SA score for Seq3, Seq7, and Seq11, it is limited in terms of handling occlusions and out-of-view situations, as demonstrated by Seq1, Seq2, Seq4, Seq6, and Seq8. Occlusion and fully out-of-view cases are generally major problems in object tracking tasks. If the tracker does not employ a strategy to update the model, the tracker may easily drift from the correct target to various distractors. To prove that the proposed method can handle such distractors appropriately, the representative quantitative results of the proposed method are compared to those of SiamRPN++ in terms of IoU overlap in Figure 5. In Seq2, the object left the view and became occluded. The proposed and SiamRPN++ methods encounter the problem of drifting to the background after frame 195. As a result, the SiamRPN++ lost the target. However, the proposed method could recover the track in frame 225 by the detection mechanism. Figure 5 shows that the proposed method maintains the tracking after the occurrence of occlusion in frame 435, by taking advantage of the redetection process.

Table 4 compares the processing time per frame in FPS of the proposed approach to that of the MedianFlow, YOLOv4, and LC methods, in order to evaluate its computational complexity. The reported speed was obtained using an Intel i7-6700 CPU, and the YOLOv4 result was obtained using the Intel i7-6850K CPU and RTX 2080 SUPER GPU. In this experiment, the YOLOv4 is performed independently; hence, the speed of the proposed and LC methods was measured, except for the performance of YOLOv4. MedianFlow, which is a light model that achieves tracking by estimating the displacement of a number of points within the bounding box, ran at 40.84 FPS. Even though the speed of the LC method decreases by 84% of MedianFlow’s, it still ran at an effective real-time speed of 34.35 FPS. YOLOv4 achieved the best result of 49.46 FPS; however, this method suffers a critical limitation in terms of distinguishing the target object from other objects in the same class. As shown in Table 4, the proposed method obtained a comparably inferior speed of 21.90 FPS; however, the proposed method can run in effective real-time on applications that require greater than 20 FPS. In addition, the proposed method has demonstrated outstanding robustness; thus, it could offer a reasonable trade-off between accuracy and speed by adjusting the number of weak classifiers of boosting and the detector execution period.

#### 4.3.2. Evaluation for Object Detection

The proposed method was primarily designed for object tracking; however, it can also be applied to object detection. In reference to the literature [50,51], we tested the proposed method compared to the YOLOv4 and LC methods in terms of precision, recall, and F-measure.

As shown in Table 5, the proposed method outperformed the compared methods in terms of TP, recall, and F-measure. YOLOv4 predicts bounding boxes by exploiting a pretrained model on vast datasets. In this evaluation, YOLOv4 achieved the best precision value (0.930) derived from the lowest FP. The proposed and LC methods can suffer from drifting, as demonstrated by the FP results; however, by using a tracking algorithm, the proposed and LC methods exhibit dominant performance in terms of FN by exploiting the benefits of both the tracker and detector. Thus, the proposed method achieved high precision (0.836) following YOLOv4, and the best recall value. In other words, the proposed method can maintain steady tracking track while not losing the target. Relative to the precision and recall results, the proposed method achieved significant F-measure performance (0.879), which proves that the detector can improve accuracy by incorporating an object tracking algorithm.

#### 4.3.3. Visual Comparison

A tracker must be able to handle various distractors to realize robust tracking in real-world applications. Figure 6 shows a visual comparison of the results obtained by the proposed method and other methods, i.e., MedianFlow, SiamRPN++, YOLOv4, and LC, on the test dataset. Note that all methods were initialized in the first frame for this evaluation.

In the first row, Seq4 demonstrates that MedianFlow and SiamRPN++ can fail to track the target in out-of-view cases. Here, as the target disappears, the trackers (except MedianFlow) identify the target’s absence, but MedianFlow drifts to the background in frame 1319. Then, when the target is revealed, the proposed, LC, and YOLOv4 methods can locate the target accurately with more reasonable localization confidence due to the use of the detection mechanism. These results demonstrate that using only a single tracker, e.g., SiamRPN++, cannot realize long-term visual tracking in out-of-view cases without employing a redetection mechanism.

In Seq5, the proposed, LC, and SiamRPN++ methods quickly adapted to the fast motion and great scale variations of the target, despite the low resolution, while MedianFlow and YOLOv4 failed to realize the same adaptation. This result proves that the YOLOv4 method suffers when handling small objects.

Seq6 illustrates the behavior of the methods in the out-of-view case caused by rapid camera movement. After frame 162, MedianFlow estimated the oversized bounding box compared with the ground truth. In this case, SiamRPN++’s target drifted to the background; however, it recovered the tracking after the target appeared in the search area. Note that the proposed and LC methods recognized the target’s absence and maintained tracking from the position at which the YOLOv4 method detected the target’s reappearance.

Finally, the results obtained for Seq7 demonstrate that the target became occluded by a building structure and then moved behind it. In addition, several distractors of similar flying objects frequently appeared in this case. As shown in Figure 6, from the beginning, MedianFlow gets confused while tracking the tiny target (here, the target was only approximately 80 pixels). YOLOv4 failed to detect the target by predicting a another similar object as a drone; hence, the LC method failed to track because it is incapable of recognizing the changes in the target’s appearance. In contrast, the proposed method was able to locate the target accurately by handling incorrect detection results, using the integration mechanism that exploits the online learned classifier.

Thus, we believe that the proposed method demonstrates beneficial use of the online updating mechanism and the combined detection and tracking method framework.

### 4.4. Evaluation on Other Datasets

We evaluate the proposed method on datasets commonly used for aerial tracking, such as UAV123 [55] and UAVL [55]. The UAV123 and UAVL datasets include low-altitude aerial videos captured from a UAV. The datasets reflect scenarios of common visual tracking challenges, such as aspect ratio change (ARC), background clutter (BC), fast motion (FM), full occlusion (FOC), partial occlusion (POC), illumination variation (IV), low resolution (LR), out-of-view (OV), similar objects (SOB), scale variation (SV), and viewpoint change (VC). These datasets are employed to assess whether a tracker is suitable in real-world scenarios [34].

As same as the evaluation in Section 4.3, we compare the proposed method with YOLOv4, MedianFlow, and the aforementioned integration method called LC, in terms of success and precision plots. The integration methods of the proposed method and LC employ YOLOv4 and MedianFlow as the detector and tracker, respectively. In the UAV123 and UAVL experiments, the model of YOLOv4 was trained with the COCO dataset [56]. In accordance with classes on the COCO dataset, we use 77 sequences from the UAV123 dataset and 14 sequences that include cars, trucks, boats, bikes, persons, birds, and aerial vehicles classes, except building and group classes.

#### 4.4.1. Results on UAV123 Dataset

Figure 7 shows the success and precision plots on the UAV123 dataset. The proposed method shows the best success score and second-best precision score. Figure 8 shows the success plots of the proposed and other methods for the 12 scenarios. The proposed method achieves the best success scores in 8 out of 12 scenarios, including SV, ARC, POC, CM, FM, LR, IV, and SOB. The proposed method dominantly outperforms in FM, LR, IV, and SOB scenarios.

The reason for the high success score under SV, ARC, IV, and POC includes that the target’s appearance changes are treated well by the proposed online learning. Our online learned classifier also allows handling SOB. As shown in FM and CM, the proposed method’s detector helps to re-track when the target is missing. Our tracking under LR, thanks to the MedianFlow, is suitable for tiny object tracking. This comparison shows that the proposed method takes advantage of the detector and the tracker. On the other hand, the performance of the proposed method is slightly less than the YOLOv4 or LC method in FOC, OV, BC, and VC, likely because model updating is often accompanied by a dilemma. Note that a model may not catch the appearance changes without online updating, but the updating by an appearance that is not the intended target could bring the risk of encountering the problem of drifting.

#### 4.4.2. Results on UAVL Dataset

Figure 9 and Figure 10 illustrate the experimental results on the UAVL dataset. The proposed method achieves promising results in 10 out of 12 scenarios, including FOC, POC, IV, VC, CM, SV, ARC, LR, FM, and SOB. Among other things, the proposed method shows outstanding results in SV, ARC, LR, FM, and SOB. They reveal that the proposed method also works well in long-term tracking, since it is able to track again with the help of the detector even if tracking drifts.

### 4.5. Limitations

The proposed hybrid UAV tracking method, while effective in many scenarios, still has limitations. In scenes with a high density of dynamic and tiny objects, the detection and tracking methods may fail depending on their respective abilities. Wrong estimation of the detection and tracking methods may also contaminate the online-learned classifier. Once the classifier’s ability for discrimination is weakened, there is an increased risk of the proposed hybrid tracking method drifting to the background, whether semantic or not. Another real-world application concern is processing time, since the proposed framework requires detection and tracking mechanisms. Therefore, implementing multi-threading could help reduce time consumption.

## 5. Conclusions

This paper has proposed a hybrid UAV tracking method that combines detection and tracking techniques to handle common distractors, by exploiting an online classifier-based integration method. In the proposed method, the classifier updates features in an online manner; thus, the method is robust against object deformations, e.g., appearance changes and scale variations.

To evaluate the proposed method, we constructed a custom dataset containing images of UAVs flying in outdoor environments. In addition, we considered a public drone-vs-bird dataset to train and test the tracking methods. The experimental results demonstrate that the proposed method is efficient and accurate. The proposed hybrid method with the YOLOv4 and MedianFlow techniques has demonstrated the importance of the online updating mechanism and combined detection and tracking method framework. This framework can be applied to other computer vision tasks, e.g., object detection, depending on their goals. As the proposed framework can be applied with any detection and tracking techniques, the improvement of object detection and tracking is expected to improve our method. If the detection and tracking algorithms integrated in the proposed framework have a very limited performance, our method may also be subject to their performance restriction.

## Figures and Tables

**Figure 1 sensors-23-03270-f001:**
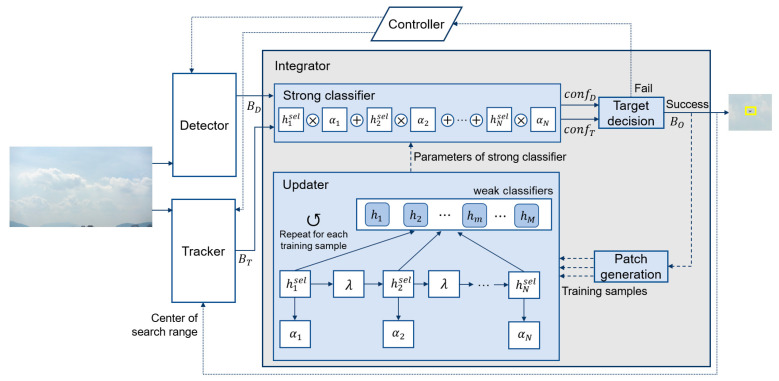
Framework of the proposed hybrid tracking method.

**Figure 2 sensors-23-03270-f002:**
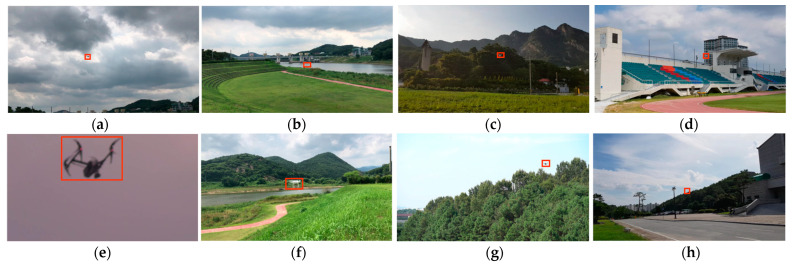
Screenshots taken from our dataset: (**a**) cloud background, (**b**) river background, (**c**) mountain background, (**d**) building background, (**e**) large target, (**f**) medium target, (**g**) small target, and (**h**) tiny target.

**Figure 3 sensors-23-03270-f003:**
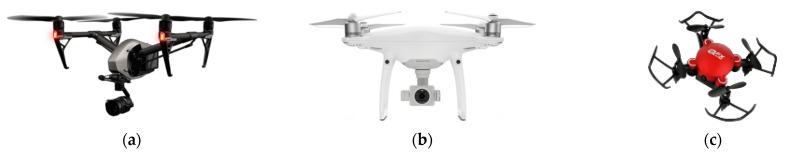
Examples of UAVs used to capture our dataset: (**a**) large UAV, (**b**) normal UAV, and (**c**) tiny UAV.

**Figure 4 sensors-23-03270-f004:**
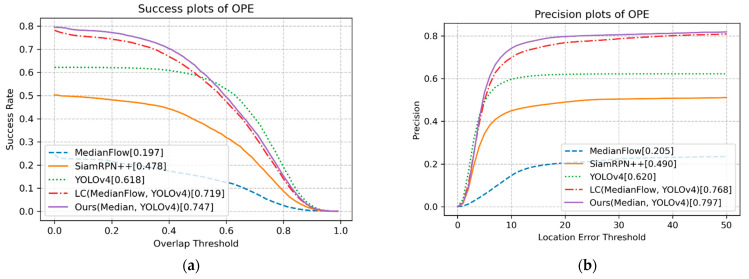
(**a**) Success plots of test sequences; (**b**) precision plots of test sequences.

**Figure 5 sensors-23-03270-f005:**
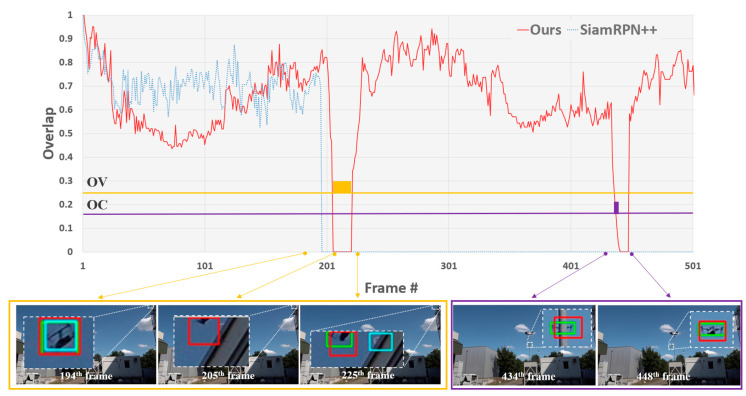
Frame-by-frame comparison of overlap score on Seq2. Yellow and purple shaded boxes indicate OV and OC, respectively. In the bottom, the red, light blue, and green boxes denote the proposed method (MedianFlow, YOLOv4), SiamRPN++ [30], and ground truth, respectively.

**Figure 6 sensors-23-03270-f006:**
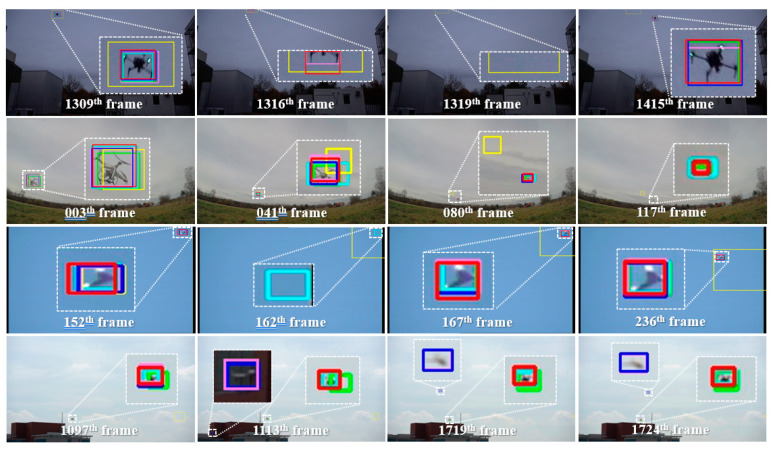
Visual results of the compared methods (from top to bottom) for Seq4, Seq5, Seq6, and Seq8. Red boxes represent the proposed method (MedianFlow and YOLOv4), green boxes represent the ground truth, yellow boxes represent MedianFlow [39], pink boxes represent YOLOv4 [20], light-blue boxes represent SiamRPN++ [30], and blue boxes represent the LC (MedianFlow, YOLOv4) method [5].

**Figure 7 sensors-23-03270-f007:**
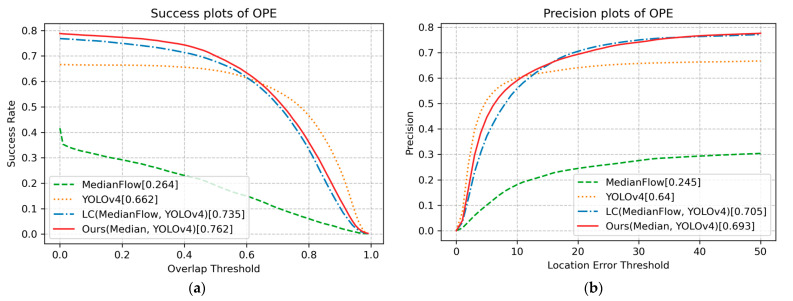
(**a**) Success plots on UAV123 dataset; (**b**) precision plots on UAV123 dataset.

**Figure 8 sensors-23-03270-f008:**
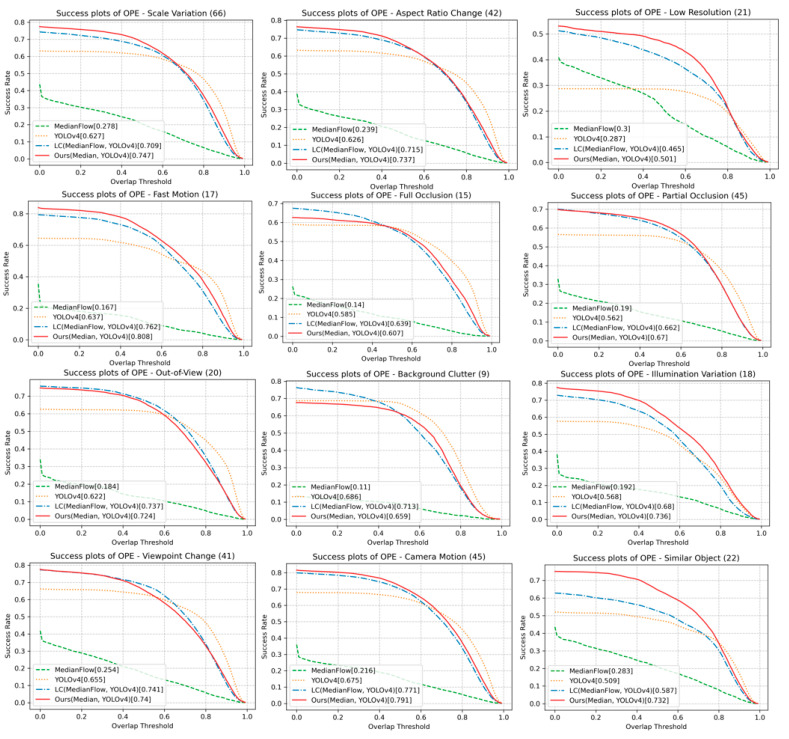
Success plots on different scenarios in UAV123 dataset.

**Figure 9 sensors-23-03270-f009:**
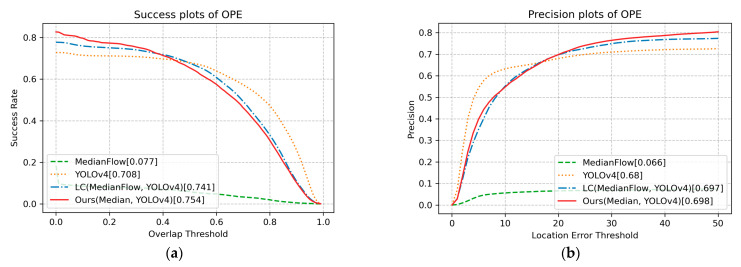
(**a**) Success plots on UAVL dataset; (**b**) precision plots on UAVL dataset.

**Figure 10 sensors-23-03270-f010:**
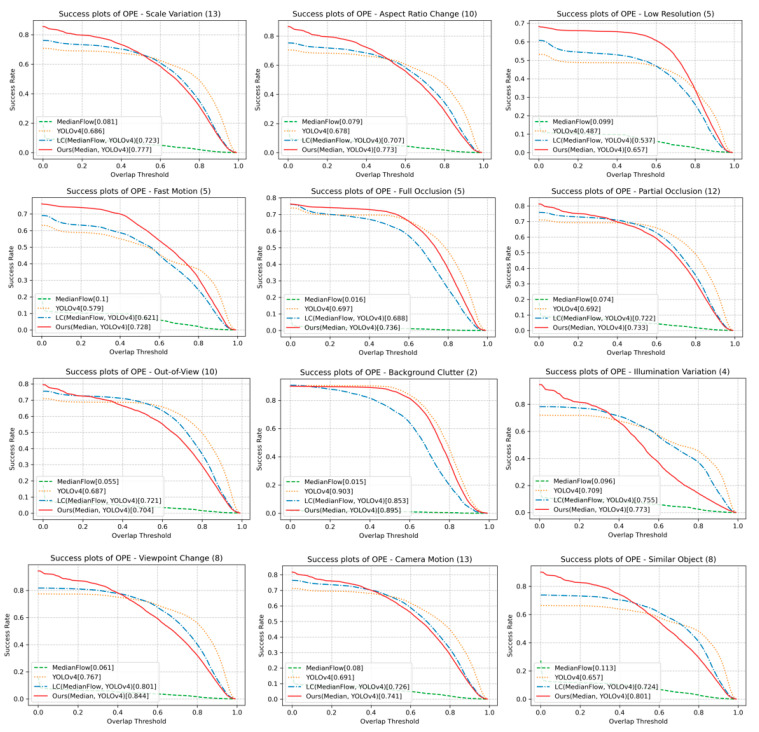
Success plots on different scenarios in UAVL dataset.

**Table 1 sensors-23-03270-t001:** Attribute annotation in dataset.

Attribute	Description
BC	Background clutter (the background has a similar color as the target or the background has changed)
CM	Camera motion (the camera is moving)
FM	Fast motion (the ground truth’s motion between two adjacent frames is greater than 60 pixels)
LR	Low resolution (the number of pixels inside the ground truth is less than 400 pixels)
OC	Occlusion (the target it partially or heavily occluded)
OV	Out-of-view (the target leaves the view)
SV	Scale variation (the ratio of the bounding boxes of the first and current frames is out of range $\left[0.4,2\right]$)

**Table 2 sensors-23-03270-t002:** Test dataset.

Dataset	Sequences	Resolution	# of Frames	Attributes (see Table 1)
Drone-vs-bird dataset [44]	Seq1 (2019_08_19_C0001_5319_phantom)	3840 × 2160	2951	OC, OV, BC
	Seq2 (2019_09_02_C0002_2527_inspire)	3840 × 2160	502	OC, OV, BC
	Seq3 (2019_10_16_C0003_5043_mavic)	3840 × 2160	426	CM
	Seq4 (2019_11_14_C0001_3922_matrice)	3840 × 2160	2601	OC, OV, LR
	Seq5 (gopro_004)	1920 × 1080	751	OC, BC, LR, CM
	Seq6 (parrot_disco_midrange_cross)	720 × 576	3001	OC, OV, BC, CM
Our dataset	Seq7	1920 × 1080	2630	OC, OV, LR, CM
	Seq8	2048 × 1536	5807	OC, OV, LR, CM
	Seq9	2048 × 1536	2701	OC, OV, CM
	Seq10	1920 × 1080	1937	CV, LR
	Seq11	2048 × 1536	1369	BC, CM
	Seq12	2048 × 1536	3834	OV, LR, CM

**Table 3 sensors-23-03270-t003:** Performance SA (%) and mSA (%) on test sequences (first-, second-, and third-ranked trackers are labeled in red, blue, and **bold**, respectively).

Tracker	SA												mSA
	Seq1	Seq2	Seq3	Seq4	Seq5	Seq6	Seq7	Seq8	Seq9	Seq10	Seq11	Seq12	
SiamRPN++ [30]	0.108	0.277	**0.856**	0.380	**0.341**	0.129	**0.507**	0.115	0.755	0.710	**0.396**	0.111	0.390
MedianFlow [39]	0.006	0.329	0.067	0.353	0.020	0.047	0.107	0.028	0.586	0.033	0.048	0.087	0.143
YOLOv4 [20]	**0.447**	**0.362**	**0.854**	**0.513**	0.340	**0.709**	**0.404**	**0.136**	**0.780**	0.776	0.107	**0.447**	**0.490**
LC [5] (MedianFlow, YOLOv4)	**0.5**	**0.455**	0.774	**0.545**	**0.468**	**0.639**	0.398	**0.238**	**0.802**	**0.713**	**0.256**	**0.467**	**0.521**
Proposed method (MedianFlow, YOLOv4)	**0.46**	**0.637**	**0.839**	**0.581**	**0.463**	**0.641**	**0.445**	**0.286**	**0.777**	**0.762**	**0.264**	**0.475**	**0.553**

**Table 4 sensors-23-03270-t004:** Comparison of FPS of methods on constructed dataset.

Sequences	MedianFlow [39]	LC [5]	Proposed Method	YOLOv4 [20]
Seq1	15.27	13.46	11.65	49.74
Seq2	14.72	12.90	10.32	49.90
Seq3	15.38	12.90	9.97	49.77
Seq4	14.24	14.03	10.85	49.81
Seq5	51.93	49.11	28.01	49.23
Seq6	89.43	81.67	43.14	48.39
Seq7	53.81	48.34	23.15	50.40
Seq8	37.92	36.65	21.06	49.08
Seq9	42.72	33.23	23.97	49.13
Seq10	53.66	44.86	30.61	49.18
Seq11	49.95	31.96	25.59	49.58
Seq12	51.09	33.09	24.52	49.38
Average	40.84	34.35	21.90	49.46

**Table 5 sensors-23-03270-t005:** Comparison of precision, recall, and F-measure on test sequences.

Tracker	TP (%)	FP (%)	FN (%)	Precision	Recall	F-Measure
YOLOv4 [20]	61.77	3.69	30.25	0.930	0.700	0.772
LC [5] (MedianFlow, YOLOv4)	71.93	21.48	5.35	0.822	0.936	0.868
Ours (MedianFlow, YOLOv4)	74.77	18.59	5.40	0.836	0.939	0.879

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
