# Peer review of "Online Learning-Based Hybrid Tracking Method for Unmanned Aerial Vehicles"

_sensors, 2023, doi:10.3390/s23063270_

Round 1

Reviewer 1 Report

The authors of the manuscript present a framework for tracking UAVs. The framework has two major highlights: (1) online learning and (2) hybrid tracking. The online learning component updates the object's features while tracking thereby allowing for robust tracking in the presence of real-time changes to the background. The 'hybrid' part combines results from the detector and the tracker to come up with an 'objectness score". The presented framework was evaluated on two separate datatsets in addition to others datasets for object detection, in order to assess its generalizability. Promising results are reported for all the datasets and the results are compared to selected published works reporting state-of-the-art results on the task.

The topic is interesting and relevant to the journal and the paper is, generally, well-written. Adequate experimentation is carried out and the results have been analyzed and compared to other published works. However, the authors can consider the following comments to further improve the quality of the paper:
1. The authors discuss the YOLO framework, moving from YOLOv1 to YOLOv4. Why not later versions of YOLO like YOLOv5/6/7?
2. In the methodology, how is the parameter N decided? Is this 'N' somehow related to the 'n' as discussed in "... every n frames..."
3. The authors presents the algorithm in Figure 2. Is the algorithm completely new or based on online learning-based adabost? In this case a reference should be provided.
4. The authors should explain line 17 of the algorithm. In general, it would be better to have comments in the algorithm for better readability and understandability.
5. The authors mention "...and voting weights for each training sample", is it for training sample or classifier?
6. The authors mention constructing a dataset, will the dataset made public so others can benchmark the results? It should be indicated in the manuscript.
7. the authors mention "In this study, the IoU threshold was set to 0.3.", the threshold seems to be quite low! Are other works using the same threshold? Are the ones compared in the Tables using the same threshold?
8. Different tables show comparisons with a selected other published works. How were these papers selected? Are they representative/state-of-the-art? A justification on the choice should be provided. For example, in Table 5 only 2 other works are selected.
9. The authors should clearly list down the strength and limitations of their work towards the end of Section 4 after presenting their results.
10. Although the manuscript is well-written, there are few mistakes in grammar and sentence structures. It is recommended that the authors proofread the entire manuscript and correct those mistakes. Examples include:
- the deep learning based-detector -> the deep learning-based detector
- in recent years, it still faces several challenges with distractors caused by similar objects and appearance changes of the tracked object [3, 4]. -> {consider rewriting}
- due to the limited among of available
- in every 30 frames or in cases -> in every 30 frames, or in cases
- the detector performs in every n frames -> performs what?
- image per 1 s to... -> image per second to...
- 4.2.1. Subsubsection -> {the Subsubsection should have a title}
- CLE score is less a given threshold -> CLE score is less than a given threshold
- because of incapable of recognizing

Reviewer 2 Report

- The content of the abstract should be increased. 

-         The authors should polish the paper suitably. The whole paper should be reviewed carefully, in order to correct all the typing errors.

-         In introduction, it is not enough to state the current work. It should be expended and reconstructed. Including the motivation, the main difficulties, the main work and the improvements compared with previous related works should be emphasized in this section.

-         There are no numerical simulation results, which can demonstrate the superiority of the proposed control law. I suggest comparing the simulations with the results of the recent (2021-2022) related valid references.

-         The novelty of the proposed method should be highlighted carefully. -         Some of the symbols are not seen correctly in the paper.

-         All of the variables and parameters should be defined in the manuscript. -         The importance of the problem considered in this paper should be further addressed. 

-     Some of the researches on quadrotor UAVs such as the following references should be studied in literature review: 

Adaptive finite-time backstepping control tracker for quadrotor UAV with model uncertainty and external disturbance

A method for autonomous collision-free navigation of a quadrotor UAV in unknown tunnel-like environments

Data fusion fault tolerant strategy for a quadrotor UAV under sensors and software faults

Adaptive barrier fast terminal sliding mode actuator fault tolerant control approach for quadrotor UAVs

SPT-Based Composite Hierarchical Antidisturbance Control Applied to a Quadrotor UAV

Fixed-time filtered adaptive parameter estimation and attitude control for quadrotor UAVs

-         No issue regarding complexity of the proposed method has been presented.

According to my comments to the paper, I recommend revision of this paper.

Round 2

Reviewer 2 Report

I have no more comments.